# Progress in Almond Quality and Sensory Assessment: An Overview

**Riccardo Massantini and Maria Teresa Frangipane ***

Department for Innovation in Biological, Agri-Food and Forestry Systems (DIBAF), University of Tuscia, via San Camillo De Lellis, 01100 Viterbo, Italy; massanti@unitus.it
* Correspondence: mtfrangi@unitus.it;

**Abstract:** World production of shelled almonds has reached 3.2 million tonnes (FAO 2020). Almond production has grown during the last decennium, also because of the consumer conviction that almonds have significant health benefits. Almonds have exceptional nutritional and organoleptic characteristics, and proper assessment of the quality of almonds is of utmost importance. Almonds have a nutritional value that is relatively low in total sugars (4.35 g/100 g of almonds) but rich in lipids, proteins, minerals, vitamins, and phytonutrients, making them a healthy and nutritious food. The almond kernel is particularly rich in protein, the second most important fraction after the lipid fraction. The protein content of almond kernel depends on the cultivar and varies from 8.4% to 35.1%. This review examines current advancements in the quality assessment of almonds, evidencing above all their nutritional characteristics, health benefits and the influence of processing on shelf life. Our aim was to provide an overview in order to improve the quality of almonds and the sustainability of the whole production. According to the literature, almonds can provide many health benefits and are a great economic resource. This review will help almond producers to choose the best cultivars to cultivate and, in the final analysis, enhance the qualitative characteristics of almonds. Our review is also an important resource for scientists. It provides state of the art research and can offer inspiration for other researchers.

**Keywords:** almond; quality; antioxidants; chemical composition; sensory analysis

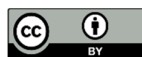

## 1. Introduction

The almond tree (*Prunus dulcis* (Mill.) D.A. Webb) is a species of tree included in the subgenus Amygdalus [1]. The almond tree is a member of the genus *Prunus* L. within the family Rosaceae, native to south-central Asia and cultivated in Mediterranean-type climates [2]. In addition to commercially cultivated almonds, there are about 30 species of wild almonds that are generally more bitter than the cultivated varieties [3]. The origin of the almond occurred ~5.88 million years ago [4]. Almond trees were originally planted as genetically diverse orchards represented mainly by bitter seedlings. Subsequently, [5] a genetic mutation controlling the sweetness of the kernel was discovered, allowing the domestication of almond as a food crop. The cultivated sweet almond *Prunus dulcis* (Figure 1) was likely selected from prehistoric populations by interspecific hybridization for their more desirable taste [6], and recently, redomesticated almond germplasm has been identified to improve nutritional qualities and food safety [7]. Almonds were one of the first domesticated fruit trees. As early as the Early Bronze Age (3000–2000 BC), domesticated almonds appeared in archaeological sites in Numeira (Jordan) [8]. Tutankhamun's tomb in Egypt (ca. 1325 BC) is another

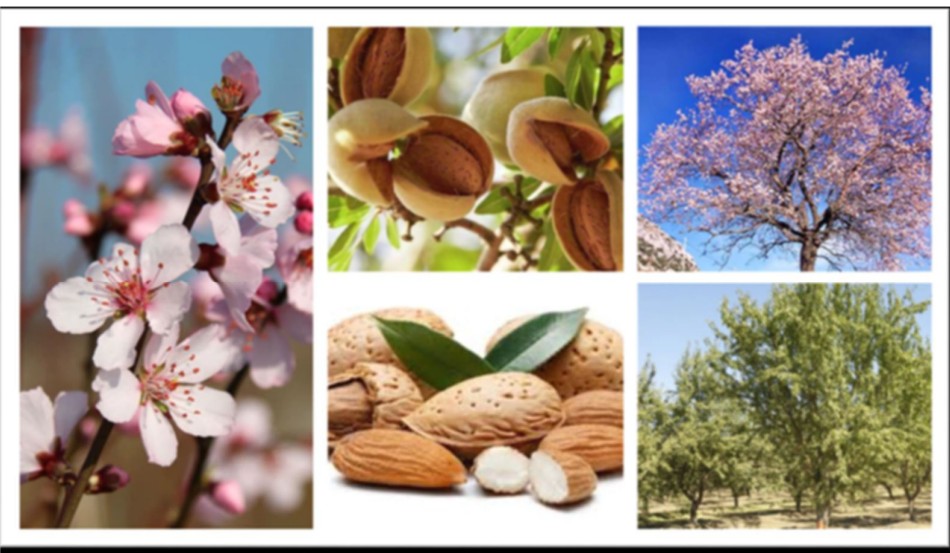

**Figure 1.** Photographs of *Prunus dulcis* trees, blooms and fruits.

Almond Lifecycle:

November through January: almond trees go through a period of dormancy, losing their leaves.

Between mid-February and mid-March: almond trees have flowering with white and light-pink blooms.

March through June: almond kernels mature and grow protected by an outer hull.

July: almond hulls begin to open.

August through October: almond hulls, fully opened, are harvested.

Well-known archaeological example of the almond fruit [9]. Delplancke et al. [10] suggested that the current distribution of *Prunus dulcis* results from human-driven spreads with an eastern Mediterranean origin. In addition, the seed almond tree has long been cultivated in the Mediterranean, at least for the last 2500 years, and was spread by humans to the western Mediterranean region. Although it originated in the Middle East, the almond tree has been cultivated more widely, including in southern Australia and California [11]. The almond tree is known by different local names in different regions of the world. It is known as sweet almond (English), lauzulhulu (Arabic), almendro (Spanish), amande (French), amygdalia (Greek) and mandorlo (Italy). For a long time, almonds there have been used for therapeutic purposes. The use of sweet almond oil for massage therapy and in Greco-Persian medicine is well known [12]. The almond tree is the first deciduous tree to flower, usually in February; it has, therefore, been associated with the rebirth of nature and imperishable hope, giving rise to myths related to its early flowering. The origin of the almond tree is linked to the Greek myth of Phyllis and Acamantus. The myth tells that Phyllis, a princess of Thrace, fell in love with Acamantus, (or Demophon as other sources say), who left to go to Troy to fight in the war. The princess promised to wait for him until the conflict was over. However, he still had not returned after more than ten years. So, Phyllis wept herself to death. The goddess Athena, moved by this sad story, transformed Phyllis into an almond tree. When Acamantus, still alive, returned from Thrace, he embraced the almond tree with all his love, which made its delicate flowers bloom (Ovid, *Remedia amoris* 591–604). Almond is the world's most important nut in terms of commercial production. According to the Statistical Database of the Food and Agriculture Organization of the United Nations (FAOSTAT), world production has reached 3.2 million tonnes of unshelled almonds. The countries of the Mediterranean basin are the main producers; however, the United States, with 1.9 million tonnes, accounts for more than 65% of world production [13]. Almonds account for 30% of the estimated

total consumption of nuts worldwide, followed by walnuts with 20%. After Europe, which is the largest consumer, North America and Asia are the second and third largest consuming regions with similar production [14]. In the United States, sweet almonds are the most consumed nut (1.03 kg per capita per year), exceeding the consumption of walnuts, the second most consumed nut, by four-fold. The economic importance is linked not only to the high quantity of production but also to the higher prices per kilogram. In this sense, the Mediterranean countries, such as Italy and Spain, have important production regions that, thanks to their mainly traditional cultivation methods, make it possible to obtain a higher quality product that can, therefore, command higher prices on the market. In fact, the price of almonds from the Mediterranean regions is around 10 dollars/kg, while Californian almonds cost about 5 dollars/kg. Almonds are the most expensive nut crop in the world. The value of supply, estimated as production times its average monthly unit price, is about $9 billion [14]. The commercial importance of the almond fruit is still related to the United States, which continues to dominate world exports of almonds. In 2019, 566,000 tons of shelled almonds were consigned to Spain (17%), Germany (8%) and Japan (7% each). Regarding Spain, shelled almond shipments were principally earmarked for European Union countries (85%) such as Germany, Italy and France. China (50%) and the European Union (31%) were Australia's top shelled almond destinations [14].

There have been several reviews on almond fruit chemical composition, nutrients and their influence on cardiovascular health [15–18]; to date, no review that investigates recent progress in assessing the quality of almonds has been carried out, so this is the goal of our review. This review includes current advancements in the quality assessment of almonds, with a focus on their nutritional characteristics, health benefits, and the impact of processing on shelf life. Our review is also an important resource for scientists. It provides a state of art offering of research and can offer inspiration to other researchers. Ultimately, this review will help almond producers improve the quality of the almonds they produce. Data in the literature, in fact, show that the influence of processing on almond quality is very strong.

## 2. Almond Fruit Characteristics

The almond tree needs a Mediterranean climate with slightly warm summers (30–35 °C) and cool winters. The unopened flowers are known to withstand cold down to −2 °C, but flowers at the petal drop stage can only withstand temperatures of −2.2 to −3.3 °C for a short period [19]. For almond cultivation, deep, loamy and well-drained soils are ideal, but medium soils may be suitable if supplemented with adequate irrigation. In this regard, the Mediterranean basin is characterized by scarce rainfall that makes irrigation interventions necessary during the period of growth of the almond tree [20]. Doll [21] argues in his study that the growth cycle of almond can be divided into several phases. Phase I corresponds to flowering and growth of fruit size, Phase II to growth of embryo size, Phase III to increase in seed weight or kernel filling. The author explained that water stress at any time during the growing season is not recommended because it will reduce vegetative growth and future almond quality. During Phase I, when the fruit and kernels are smaller, severe water stress could be the cause. Water stress during Phase II will decrease carbohydrates and thus core size. If water stress occurs in Phase III of fruit development, kernel carbohydrates will be reduced since the accelerated hull cleavage leads to a decrease in kernel dry weight. Only in Phase III, water stress is reported to have very little impact on kernel yield and quality. A recent review [22] focused on the effects of deficit irrigation strategies on the almond crop, yield and fruit quality. The concept of water sustainability involved the use of mild levels of water deficit that, with minimal yield losses, result in high-quality products. All of the information gathered in this review validates the possibility of producing high-quality almonds by reducing water consumption at specific growth stages and respecting natural ecosystems. These results were confirmed by Lipan et al. [23], who found positive correlations with dry weight, color coordinates (L*, a* and b*), minerals (K, Fe and Zn), organic acids (citric acid), sugars (sucrose, fructose and total sugars), antioxidant activity and fatty acids (linoleic, polyunsaturated and

polyunsaturated:monounsaturated fatty acids), while water stress in almonds was negatively correlated with kernel yield, water activity, weight (almond, kernel and shell), size, minerals (Ca and Mg), fatty acids (oleic acids, oleic/linoleic ratio, monounsaturated and polyunsaturated:saturated) and sensory attributes (size, bitterness, astringency, benzaldehyde and woodiness). However, the authors pointed out that the agricultural sector could conserve irrigation water consumption by about 45% while obtaining high-quality products. The almond tree is both morphologically and physiologically diversified, and the genetic structure of various species was also demonstrated to be heterogeneous. As a result, almond trees show high variability in shape, size, vigor, branching, flowering and fruits. The crown of the tree is influenced by the cultivar, and in many traditional growing regions, the trunk is about 1.5 m high. Almond tree flowers are generally perfect, pentamerous and with a single pistil [1]. Botanically, the fruit of the almond tree is a drupe with a shell consisting of the pericarp and mesocarp, and it coalesces at maturity to show the endocarp or shell. The shell contains the seed or kernel, which is the commercial part of the almond crop and is characterized by traits related to the cultivar itself. Genetically, the almond represents a very variable species [24]. This aspect is related to the fact that the almond family includes numerous inter-compatible species with a huge gene pool available for breeding. Moreover, the almond tree has one advantage over other agronomic crops in that unique gene/genomic combinations can be disseminated by clonal propagation. Almond cultivar identification was for many years based on morphological characteristics. Subsequently, taking into account difficulties due to the influence of the environment, molecular markers for variability analysis and cultivar identification have been used [24]. Halász et al. [25] established the value of markers to distinguish different genetic lines and found a large gene pool available for genetic improvement of almonds. The authors studied eighty-six almond accessions of different geographical origins, ranging from Central Asia to the United States. The results showed that mutations and massive gene exchange between different wild species and cultivated *Prunus dulcis* represented a crucial component of genetic differentiation. However, it is important to emphasize that the data showed that there is no indication of a large reduction in genetic variability in almond germplasm. The standards of the International Union for the Protection of New Varieties of Plants [26] are employed to individualize all of the different almond varieties. Recent research [27] focused on pedigree analysis of 220 almond genotypes to determine the genetic structure of current breeding stocks and breeding trends over the last 50 years. The findings demonstrated that two breeding lines, based on only three cultivars (Nonpareil, Thunder and Christomorto) have dominated current breeding in the world.

### 3. Chemical Composition and Nutritional Value

In order to better evaluate the quality of almonds, knowledge of the chemical composition (Table 1) should be indispensable information. Several studies have been conducted on the chemical composition of almond fruits [28–33]. De Giorgio, Leo, Zacheo and Lamascese [28] studied the characteristics of 52 almond cultivars from the Puglia region in Italy to identify the productivity, types and quality of cultivars. The most productive cultivar was 'Barlettana', with just over 2 kg of almonds per tree, followed in order of productivity by 'Cristomorto', 'Santoro', 'Catuccia', 'Filippo Ceo', 'Piangente' and 'Pidocchioso'. The almond cultivars with the highest total lipid content, of 633 mg/g fresh weight, were the 'Filippo Ceo' varieties, while 450 mg/g was the value found for the 'Cosimo di Bari' and 'Gioa' varieties. The $\alpha$-tocopherol content was the most important parameter for classifying cultivars into groups with higher similarities as it was the parameter with the highest variability. The highest $\alpha$-tocopherol contents were obtained in the cultivars 'Senz'arte', 'Pizzuta D'Avola' and 'Rachele'. Yada, Lapsley and Huang [2] contributed to the knowledge of almond composition with a review of lipids, fatty acids, proteins, amino acids, carbohydrates, minerals and vitamins. Considerable variability in lipid content within and among different varieties was reported. The paper also showed that total lipids ranged from 25 to 66 g/100 g almonds (fresh weight). Oleic and linoleic acids

accounted for about 90% of total lipids, and saturated fatty acid levels were very low (<10%) in all varieties. Oleic/linoleic acid ratios differed widely among varieties. Total protein content ranged from 14 to 26 g/100 g of almonds. In all almond varieties evaluated, α-tocopherol was observed as the main isomer of vitamin E. Drogoudi et al. [29] determined the protein and potassium (K), phosphorus (P), calcium (Ca) and magnesium (Mg) contents of almonds in 72 cultivars and accessions grown in France, Greece and Italy to explore and enhance almond genetic resources in Europe.

**Table 1.** Chemical composition and nutritional characteristics of almonds.

| Almond Cultivars | Total Lipids Content (mg/g) | α-Tocopherol (μg/g) | Oleic/ Linoleic Ratio | Unsaturated Fatty Acids % | Carbohydrate Content (g/kg Fresh Weight) | Antioxidant Activity (μmol Trolox Equivalents Per Gram) | K (mg/ 100 g dm) | P (mg/ 100 g dm) | Mg (mg/ 100 g dm) | Ca (mg/ 100 gdm) | Total Polyphenols (mg/kg) | Protein (g/100 g) | References |
|---|---|---|---|---|---|---|---|---|---|---|---|---|---|
| 52 almond cultivars from the Apulia region of Italy | 450–633 | 218–777 | | | | | | | | | | | De Giorgio et al., 2007 [28] |
| 72 almond genotypes from France, Greece and Italy | | | | | | | 488–1235 | 310–748 | 159–321 | 206–663 | | 10–29 | Drogoudi et al., 2012 [29] |
| 7 almond cultivars from California | | 219–310 | | | | | 664–773 | 462–526 | 256–278 | 234–330 | | 20.2–22.5 | Yada et al., 2013 [34] |
| 10 genotypes and 2 commercial cultivars (Ferragnes and Ferraduel) from Turkey | | | | 90.27–92.09 | | | 679.53–986.63 | 584.37–697.31 | 225.27–381.93 | 189.63–332.19 | | 20.41–25.82 | Simsek et al., 2018 [35] |
| 10 almond cultivars from Australia, California, Italy and Spain | 423.9–561.7 | | | | 157.1–266.3 | 12.69–60.99 | | | | | 391.98–11030.53 | 14.12–22.08 | Summo et al., 2018 [30] |
| 16 almond cultivars including 12 from Tunisia, 2 from Italy, 1 from Spain and 1 from France | 477.5–609.5 | | 2.76–5.67 | 88.38–91.65 | | | | | | | | 14.49–27.15 | Gouta et al., 2020 [32] |
| 10 wild almond Iranian accessions | | | | | | | 570–890 | 430–700 | 220–400 | 210–370 | | | Zahedi et al., 2020 [33] |

Significant differences were observed between the proteins of the samples. The protein content varied by almost three-fold, ranging from 10 to 29 mg/100 g fresh weight. The results showed that potassium content predominated, with values ranging from 465 to 1235 mg/100 g dry matter, and the highest values were found in the four almond accessions from Italy, with potassium values above 958 mg/100 g dry matter. Phosphorus content ranged from 310 to 748 mg/100 g dry matter. Calcium content ranged from 160.0 to 663.0 mg/100 g dry matter, and the highest value was found in 'Truoito' from Greece. Magnesium content ranged from 159 to 334 mg/100 g dry matter. The authors found significant differences in nutrient content. This aspect is particularly important when consuming the recommended daily amount of almonds, in order to select cultivars with high nutrient content. Protein content is considered the most appropriate parameter to be used to mark differences between genotypes and to identify the origin of the almond. In

addition, the mineral and protein contents of almonds depend on the genotype rather than being related to their origin. In contrast, Yada, Huang and Lapsley [34] found that the macronutrient and micronutrient profiles obtained were overall similar for the almond varieties studied. Their research investigated the variability in nutrient composition among seven commercially important California almond varieties (Butte, Carmel, Fritz, Mission, Monterey, Nonpareil, and Sonora) collected in the northern, central, and southern growing regions of California. Among the varieties studied, Sonora showed the highest protein content with a value of 22.5 g/100 g, while the Carmen variety had the lowest, at 20.2 g/100 g. A difference of only 9.1 mg in $\alpha$-tocopherol per 100 g of almonds was observed between the varieties, with the highest content of 31 mg/100 g (Sonora) and the lowest of 21.9 mg/100 g (Monterey). The results showed significant differences in the amounts of minerals in almond farming in central, northern, and southern California. This effect was expected, since it is generally known that mineral content in plant tissues is influenced by environmental and agronomic factors including soil composition, irrigation, water sources, and fertilizer components. In fact, these minerals accumulate during growth and ripening of almond fruits. Summo et al. [30] highlighted the influence of harvest time and cultivar on the chemical composition of diverse species of almonds. Authors studied both the early stage of almond maturity and the complete maturity of fruits. Strong variations in the chemical composition of almonds were observed, with growth in lipids and a reduction in carbohydrates and proteins found. The lipid content ranged from 200.6 to 301.1 g/kg of fresh matter when the fruits were unripe and increased considerably to 561.7 g/kg when the fruits were ripe, with dry brown hull. Total phenolic compounds showed great variability among cultivars, ranging from 391.98 mg/kg to 11,030.53 mg/kg. Antioxidant activity significantly related to total phenolic content varied greatly among cultivars, ranging from the lowest value of 12.69 up to 60.99 $\mu$mol Trolox equivalents per gram. These parameters appeared to rise with harvest time. The authors observed that greater phenol amounts and fewer lipids had a positive effect on shelf life, with reduced oxidation in the storage period. Gouta et al. [32] studied the chemical and nutritional composition of 12 Tunisian almond cultivars ('Dillou', 'Khoukhi', 'Blanco', 'Abiodh', 'Lsen Asfour', 'Achaak', 'Zahaaf', 'Fekhfekh', 'Ksontini', 'Sahnoun', 'Porto', and 'Mahsouna') compared to five almond cultivars from Italy ( 'Mazetto' and 'Supernova'), Spain ('Francoli'), France ('Lauranne' and 'Fournat de Breznaud'). The data showed that the protein content, ranging from 14.49 to 27.15%, presented a similar variability compared to previous studies [29], in which the protein content varied from 10 to 29 g/100 g among the seventeen almond cultivars studied. At the same time, the authors observed the high content of unsaturated fatty acids, mainly oleic acid (65.29–76.21%), which is responsible for the increase in the phytonutrient value of almonds and associated with low levels of linoleic acid, extending the shelf life of almonds. In fact, the highest oleic/linoleic ratio (varied from 2.76 to 5.67) is considered a significant quality criterion for the preventive effect on lipid oxidation, particularly when almonds will be stored for long periods [36]. The results showed that oil, sugar and protein contents in almonds depend on both genotype and environmental effects. These results are in agreement with those reported by Zahedi, Abdelrahman, Sadat Hosseini, Yousefi and Phan Tran [33] who confirmed that chemical composition is significantly influenced by cultivar and environmental conditions. Among the investigated mineral elements, potassium and phosphorus showed the highest contents in wild almonds, reaching values of up to 890 mg/100 g dry matter for potassium and 700 mg/100 g dry matter for phosphorus. In this regard, it was reported that wild almonds showed high variations in mineral content, reflecting both the influences of environmental and genetic factors. On the other hand, areas characterized by low to moderate cumulative rainfall from January to June and clay soils showed the highest mineral levels, while areas characterized by high cumulative rainfall showed the lowest levels, due to leaching of minerals from the soil. Thus, the results indicated a significant effect of geographical location on the micromineral and macromineral contents of almonds. Fatty acid profile showed significant differences among wild almond accessions. Oleic acid had

the highest levels, followed by linoleic. These parameters are mainly dependent on different genotypes of almonds and are ascribed principally to the incidence of genotype–environment interactions. The authors consider the variation of fatty acids as a fundamental factor regarding the diverse industrial uses for almonds. These data are relatively in disagreement with those obtained by Simsek et al. [35], who showed a low variance in almond genotypes. The authors examined the fatty acid and mineral compositions of ten almond genotypes and two commercial varieties called Ferragnes and Ferraduel grown under the same ecological conditions in Turkey. The protein content ranged from 20.41 to 25.82 (g/100 g). All genotypes and varieties had protein values above 20%; it follows that they were considered to be rich in protein and could serve as dietary supplements. This fact could be explained considering that the experiment was conducted, for all samples, under the same ecological conditions around the Firat River, which are similar mainly to those of the Mediterranean region. Among the studied genotypes and cultivars, the highest potassium value was 986.63 mg/100 g, and 679.53 mg/100 g was the lowest. Phosphorus, magnesium and calcium contents ranged from 584.37 to 697.31 mg/100 g, 225.27 to 381.93 mg/100 g and 189.63 to 332.19 mg/100 g, respectively. It is generally accepted that almond minerals depend on many ecological factors and agronomic practices, including geographical location, soil composition, irrigation regime and fertilizer components. The genotypes and cultivars studied had similarly high mineral levels, probably influenced by the same ecological conditions. Regarding fatty acids, the authors found that the samples were rich in oleic and linoleic acids. Oleic acid was the predominant one with values from 69.76–72.02%, followed by linoleic acid (18.82–21.62%). In this regard, it is important to emphasize that the composition of fatty acids is related to oxidative stability and some nutritional characteristics. In fact, a higher percentage of unsaturated fatty acids reduces the risk of coronary heart disease. From this point of view, it can be concluded that the almond, due to its richness in unsaturated fatty acids, can be included in the diet to improve human health. A review of recent research on the chemical characterization of almond cultivars of different geographical origins has been carried out [31]. The contents of macronutrients, tocopherols, phytosterols, polyphenols, minerals, amino acids and volatile compounds along with DNA fingerprinting were reported as potential markers of cultivar and origin. The results showed that no single almond compound could be a biomarker to find differences among almond cultivars. As pointed out in a previous part, almonds manifest great variability in their chemical composition. It is, therefore, necessary, as in a puzzle, to select and then combine all of the variables with the application of multivariate statistical techniques. The nutritional value of almonds estimated by the United States Department of Agriculture [37] and presented in Table 2 confirmed that almonds are relatively low in total sugars (4.35 g/100 g of almonds), but are rich in lipids, proteins, minerals, vitamins, and phytonutrients that make them a healthy and nutritious food. Almond kernel is a food rich in protein, the second most important fraction after the lipid fraction. The protein content of almond kernel depends on the cultivar and varies from 8.4%, found in Spanish samples [38], to 35.1%, found in Moroccan samples [39]. These results are reported in several studies in the literature, as provided by Roncero, Álvarez-Ortí, Pardo-Giménez, Rabadán and Pardo [18] in a recent review on the non-lipid components of almond kernel, considering in particular the protein fraction, carbohydrates and mineral fraction. Regarding the protein profile, amandine, the main protein in almonds, was dominant, accounting for 65% of the total protein content. The authors also highlighted among the free amino acids the presence of glutamic acid and aspartic acid, followed by arginine.

**Table 2.** Nutritional value of 100 g almonds.

| Name | Amount | Unit |
|---|---|---|
| Water | 4.41 | g |
| Energy | 579 | kcal |
| Protein | 21.2 | g |
| Total lipids (fat) | 49.9 | g |
| Ash | 2.97 | g |
| Total sugars | 4.35 | g |
| Calcium, Ca | 269 | g |
| Iron, Fe | 3.71 | mg |
| Magnesium, Mg | 270 | mg |
| Phosphorus, P | 481 | mg |
| Potassium, K | 733 | mg |
| Zinc, Zn | 3.12 | mg |
| Selenium, Se | 4.1 | mg |
| Riboflavin | 1.14 | mg |
| Niacin | 3.62 | mg |
| Pantothenic acid | 0.471 | mg |
| Vitamin B-6 | 0.137 | mg |
| Folate, total | 44 | mg |
| Vitamin E (alpha-tocopherol) | 25.6 | mg |
| Fatty acids, total saturated | 3.8 | g |
| Fatty acids, total monounsaturated | 31.6 | g |
| Fatty acids, total polyunsaturated | 12.3 | g |
| Cholesterol | 0 | mg |
| Beta-sitosterol | 130 | mg |
| Tryptophan | 0.211 | g |
| Threonine | 0.601 | g |
| Isoleucine | 0.751 | g |
| Leucine | 1.47 | g |
| Lysine | 0.568 | g |
| Methionine | 0.157 | g |
| Cystine | 0.215 | g |
| Phenylalanine | 1.13 | g |
| Tyrosine | 0.45 | g |
| Valine | 0.855 | g |
| Arginine | 2.46 | g |
| Histidine | 0.539 | g |
| Alanine | 0.999 | g |
| Aspartic acid | 2.64 | g |
| Glutamic acid | 6.21 | g |
| Glycine | 1.43 | g |
| Proline | 0.969 | g |
| Serine | 0.912 | g |

Source: National Nutrient Database for Standard Reference (2019) United States Department of Agriculture.

Phenylalanine, alanine, serine and threonine were also present, although in smaller amounts. In almonds, the most representative carbohydrates (14–28%) were soluble sugars (mainly sucrose). The mentioned review shows the average value of the major mineral

elements in the almond kernel. Potassium was the mineral with the highest content (435–2051 mg/100 g), followed by phosphorus (119–873.8 mg/100 g), and both represented more than 70% of the total mineral fraction. In this regard, it is important to note that the authors assumed that the differences in protein content found could be related to the different analytical methods used. Indeed, a specific conversion factor of 5.18 could be used to estimate the protein content, since amandine, which is the dominant protein in almonds, is a globulin containing 19.3% nitrogen. Other studies, however, have used the general conversion factor (6.25), which could lead to overestimation of protein content.

## 4. Antioxidant Compounds

There is growing interest in almond antioxidant compounds because of their multiple functions, antioxidant and nutraceutical properties, and potential to extend the shelf life of almonds. The phenolic and tocopherol contents of 15 commercially significant almond cultivars were determined by Yildirim, A.N., Yildirim, F., Şan, Polat and Sesli [40]. They found wide variations in phenolic contents among cultivars. Among phenolic substances, the highest content was obtained for catechin and especially in cultivar 'Ferraduel', with values from 117.59 to 145.86 mg/kg dry weight. Additionally, the highest content of epicatechin was found in the cultivar 'Ferraduel' (from 21.07 to 27.57 mg/kg), while the gallic acid value ranged from 1.22 (Ferraduel cv) to 3.26 mg/kg (Nonpareil cv). The authors pointed out that the contents of ferulic acid, kaempferol, naringenin and p-coumaric acid were lower than those of the other phenolic compounds. Importantly, these phenolics were more stable. Since phenolic contents varied from year to year and the variation was significant only in some cultivars, it was concluded that some factors such as bacteria, pests, air and light could influence this difference. These data are in agreement with those reported in other studies [41,42], which also highlighted environmental traits in the growing region, cultivation techniques, fruit maturity status, soil properties and genetic traits of cultivars as responsible for significant differences. The tocopherol contents of almond cultivars were also studied, and significant differences were observed among cultivars. In particular, the highest content was obtained in $\alpha$-tocopherol, with mean values from 899.49 to 945.41 mg/kg in the cultivar 'Supernova'. That almond is the fruit with the highest tocopherol content among in-shell fruits has been confirmed by several research studies [43–45]. This characteristic ensures that the fruits can be stored for a long time. Fallico, Ballistreri, Arena and Tokusoglu [46] reported almond phenols as a mixture of flavonoids, phenolic acids and tannins that contribute to their antioxidant capacity in a synergistic manner. In this regard, it has been reported that most of the total phenols present in walnuts are contained in the peel [47]. The authors determined the total phenols, flavonoids, and phenolic acids in the skins and kernels of California almonds (Prunus dulcis) for the major almond varieties (Butte, Carmel, Fritz, Mission, Monterey, Nonpareil, Padre, and Price). Total phenols ranged from 127 (Fritz) to 241 (Padre) mg gallic acid equivalent/100 g fresh weight. They found that 60% of the almond phenols were, on average, `present in the peel. Interestingly, eight of the 19 flavonoids and three phenols were found exclusively in the peel; on average, 94% of the individual flavonoids were from the peel. This finding could be explained by the role of flavonoids as phytoalexins, which are localized in the skin layer surrounding the seeds and nuts, protecting them from bacterial, fungal and other environmental stresses. Interestingly, research [48] showed that flavonoid content and antioxidant activity of almonds depended more on cultivar than seasonal differences. In this study, the polyphenol content and antioxidant activity of Nonpareil, Carmel, Butte, Sonora, Fritz, Mission, and Monterey almond cultivars harvested in three seasons in California were examined. The average polyphenol content in seven almond cultivars ranged from 3.96 (Fritz *cv*) to 10.7 (Sonora *cv*) mg/100 g almonds. As well as their polyphenol content, Sonora *cv* had both the highest total phenol concentration of 159 mg gallic acid equivalent/100 g and the ferric antioxidant reducing power (FRAP) value of 891 μmol Trolox equivalent/100 g. The authors pointed out that since polyphenols and other antioxidant constituents may contribute to the health effect

of almonds, revealing an association with the most polyphenol-rich cultivars could be useful for their potential health benefits. This is probably due to the fact that different cultivars possess their own phenolic heritage, which characterizes them. Furthermore, in addition to the variation in polyphenol content among cultivars, the seven California almond cultivars had unique polyphenol profiles. It was also noteworthy that among the 18 polyphenols quantified by LC-MS analysis, isorhamnetin-3-O-rutinoside (Iso3R) was the predominant polyphenol among the almond cultivars (1.65 to 2.98 mg/100 g almonds). Finally, this study concluded that the flavonoid content and antioxidant activity of almonds may be more dependent on cultivar than seasonal differences. These results were previously found by [49]. Bolling [50] devoted a comprehensive study to the determination of polyphenols in almonds, summarizing the methods of analysis and evaluating the contribution of polyphenols to almond quality and health-promoting activity. The author showed that about 130 different polyphenols were identified in almonds, and the means per 100 g of almonds reported in the literature were 162 mg (67.1 to 257) proanthocyanidins (dimers or larger), 82.1 mg (72.9 to 91.5) hydrolysable tannins, 61.2 mg (13.0 to 93.8) flavonoids (non-isoflavone), 5.5 mg (5.2 to 12) phenolic acids and aldehydes, and 0.7 mg (0.5 to 0.9) isoflavones, stilbenes, and lignans. The positive health-promoting activity of almonds, such as lowering cholesterol, reducing the risk of developing type 2 diabetes, and hepatoprotective, antidepressant, memory enhancing, anti-aging and appetite control effects have been widely studied [51–56]. In particular, studies that have considered the contributions of almond polyphenols [50,57] have suggested putative effects on antioxidant function, detoxification, antiviral activity, anti-inflammatory function, and blood pressure. All in all, almonds have a polyphenol profile that contributes to both their dietary quality and health-promoting actions. Nowadays, consumer awareness that almond consumption is often associated with higher nutraceutical quality is increasing. Another important aspect is the environmental effect that influences some constituents in almond fruit, such as tocopherol. These compounds are, in fact, related to temperature and drought during fruit growth [45]. High tocopherol concentrations were found in almonds during years with high temperatures, showing that environmental factors had an influence on tocopherol synthesis during kernel development [58]. Regarding solar irradiation, it was found [59] that tocopherol increased in Nonpareil almond cv after UV radiation. Maestri et al. [60] reported that in arid Northwestern Argentina, where there are summer months with warmer mean temperatures during kernel development, tocopherol was found in high concentrations. Therefore, it is possible to conclude that drought and heat are the most important stresses affecting tocopherol content in almonds, with increased levels at higher temperatures and water deficit conditions. Recently, some researchers have pointed out that, among the possibilities of improving the quality of almonds, it is feasible to resort to deficit irrigation as a suitable strategy to enhance their nutritional value [61]. The findings showed significant effects on antioxidant activity and total phenols ($p < 0.001$) in response to irrigation. In particular, the antioxidant activity in almonds was increased by using deficit irrigation strategies that could maximize water conservation without compromising yield. Regarding total phenols, the authors highlighted a positive correlation with deficit irrigation treatment. Both antioxidant activity and total phenols are very important parameters, not only for health properties but also for their contribution to almond quality and shelf life.

## 5. Processing Influence on Almond Quality

Commonly, almonds are submitted to different industrial processes (soaking, hulling, boiling, blanching, roasting, etc.), and knowing the effect of different processing operations represents a fundamental point for qualitative enhancement. Vàzquez-Araùjo et al. [62] studied the changes in CIEL*a*b* color, volatile compounds and sensory parameters during roasting of almonds. The two most important almond cultivars grown in eastern Spain, 'Comuna' and 'Marcona', were studied. The authors found that in both almond cultivars, the brightness (L*) values decreased significantly during roasting as, due to

browning reactions, the samples became dark. On the other hand, the chroma (C*) values increased during the roasting treatment; in fact, the samples of both cultivars showed a brownish and more intense color. This evidence showed that the color coordinates were significantly affected only by roasting and not by almond cultivar. Regarding the chemical groups related to 'roasted smell', in this work a total of 18 pyrazines, 5 furans and 3 pyrroles were identified and quantified in both almond cultivars. The most abundant pyrazine was 2,5-dimethylpyrazine (0.30 mg/kg mean value for all times and cultivars studied) followed by 2-ethyl-3-methylpyrazine (mean of 0.20 mg/kg), 2-methylpyrazine (mean of 0.19 mg kg/) and 2,5-dimethyl-3-ethylpyrazine (mean of 0.09 mg/ kg). Sensory analysis showed that 23 min was too long a period for 'Comuna' almonds, as burnt notes appeared. The instrumental and sensory data suggested that the optimal roasting time at 200 °C for 'Comuna' and 'Marcona' almonds should be 20 min. Bolling, Blumberg and Chen [63] evaluated the content of total phenols, phenolic acids and antioxidant activity of the skins of California almonds subjected to roasting. The authors found that roasted almond skins had phenolic acids equivalent to raw almond skins, with values of 1537 μg/g for roasted almonds and 1557 μg/g for raw almonds. Roasted almonds had 26% less total phenolics (18.5 vs. 25.1 mg/g in roasted and raw samples, respectively) and 34% less antioxidant activity than raw skins (119 vs. 179 μmol of Trolox equivalents per gram, in roasted and raw samples, respectively). Since most commercial almonds are roasted, these results seem very important for the contribution of roasted almonds to dietary polyphenol intake. In another study [64], a method was developed to analyze the variation in volatile profile in raw and roasted almonds. A total of 58 volatiles were identified in raw and roasted almonds, and all volatiles increased with roasting except for decreases in benzaldehyde, 2-methyl-1-propanol, 3-methyl butanol, 2-phenylethyl alcohol, a-pinene and methyl-sulfanylmethane. The authors found that straight-chain aldehydes and alcohols demonstrated only minimal increases, while levels of branched-chain aldehydes, alcohols, sulfur-containing compounds, and heterocyclic compounds increased the most. This type of difference could be related to the chemical formation mechanisms of these two groups of compounds. This is particularly important because the small increases in straight-chain volatiles reflect heat-induced oxidation during roasting. The largest increases in branched-chain aldehydes, alcohols, sulfur-containing compounds, and heterocyclics are related to the Maillard reaction. They observed that benzaldehyde decreased from 2934.6 ng/g (raw almonds) to 315.8 ng/g (averaged across the evaluated roasting treatments, i.e., 28, 33 and 38 min) after roasting. Benzaldehyde is characterized by a pleasant almond aroma and results from the enzymatic breakdown of the diglucoside amygdalin [65]. As a result, benzaldehyde losses cause a reduction in almond flavor. Pyrazines were detected only in roasted almonds. The concentration of most alcohols was observed to increase significantly in the roasted samples except for 2-methyl-1-propanol, 3-methyl-1-butanol and 2-phenylethanol, which decreased by 68%, 80% and 86%, respectively. Among the volatiles, it was observed that the amounts of 2-pentylfuran were more strongly correlated with increasing roasting time. A factor very important for consumer acceptance is the kernel color, which is an indicator of the brown pigments formed during the browning and caramelization process. In this regard, a color change was reported in the Akbadem variety from the Aegean region of Turkey during roasting processes [66]. Almond kernels were roasted at three different temperatures (150, 160, and 170 °C) and using four roasting times (10, 20, 30, and 40 min). The results showed that darker almond kernel color was related to increases in roasting temperature and roasting time. The authors highlighted L, a, and b values as the three dimensions of the measured color. The L-value has a range from 0 (black) to 100 (white). The a- value represents the green ± red spectrum with a range from –60 (green) to +60 (red). The b- value represents the blue ± yellow spectrum with a range from –60 (blue) to +60 (yellow) [66]. The L values of samples roasted at 150, 160 and 170 °C for 40 min decreased to 52.34, 47.96 and 43.17, respectively. On the other hand, the a- and b- values increased depending on the increase in the roasting temperature and roasting time, with a-values of 9.52, 11.96 and 14.25, respectively, for samples roasted

at 150, 160 and 170 °C for 40 min; b values showed an increase to 24.25, 28.71, and 29.53 respectively. These findings agreed with those of Kaftan [67], indicating both pigment destruction and Maillard browning. It is hypothesized that the change in color during thermal processing takes place via different mechanisms, including the degradation of pigments, oxidation of ascorbic acid, and the Maillard reaction. Regard sensory analysis, the lowest scores were obtained for the almond samples roasted at 170 °C for 40 min. In fact, the panelists gave low scores for taste (1.9), color (2.1) and flavor (2.4) to the samples roasted at 170 °C for 40 min. This evidence points out that almond color may be used as an important quality indicator. The analysis of literature evaluating the effect of processing on almond color showed that if the appropriate roasting temperatures and times are selected, high-quality almond color, taste, and flavor characteristics may be obtained. This has been reported by Makinde and Oladunni [68], who analyzed the effects of processing treatments on the nutritional quality of tropical almond *(Terminalia catappa* L.). Precisely, the authors evaluated the proximate compositions, mineral, vitamin and antinutritional concentrations in almond nut under different processing methods (soaking, blanching, autoclaving and roasting). Among the considered treatments, roasting at 160 °C using different roasting times of 5,10 and 15 min greatly influenced the nutrient and antinutrient composition of almond kernels. Roasting for 15 min caused a significant increase in potassium (9.87 vs. 13.94 mg/100 g), calcium (4.66 vs. 6.76 mg/100 g) phosphorus (5.48 vs. 7.85 mg/100 g) and magnesium (4.45 vs. 6.39 mg/100 g), compared to raw kernels. As noted, mineral amounts were increased by roasting for 15 min, resulting in the highest increase in potassium, calcium, phosphorus and magnesium (by 41.2, 45.1, 43.3 and 43.6 percent, respectively). This increase in mineral contents of the samples compared to raw kernels could be due to the lower concentrations of antinutrients. In fact, the concentrations of some antinutrients in roasted samples had the largest reduction, which may be due to insoluble phytins formed between phytate and some minerals. Among antinutrients, the values for oxalate were 0.15 mg/100 g and 0.01 mg/100 g for raw and roasted almond kernels, respectively. This is due to the fact that oxalates are water-soluble, and that processing time had a great effect on oxalate concentrations in the kernels. It is known that oxalates have the ability to form water-soluble salts by binding to minerals such as sodium or potassium. Therefore, due to the solubility of oxalate in water, processing, such as boiling and roasting, allows the content of these compounds to be reduced considerably. Moreover, the decrease in oxalate could be explained considering that these compounds are thermolabile. As an overall consideration of this research, roasting at 60 °C for 15 min appears to be the recommended processing method to maintain high nutritive almond value since it provided an appreciable amount of minerals and the largest reduction in antinutrients (phytate, oxalate). In a recent work [69] the effects of roasting on nutritive value (fatty acid composition), sensorial characteristics and bioactive compounds of four Portuguese almond cultivars (Casanova, Molar, Pegarinhos and Refêgo) and two foreign cultivars (Ferragnès and Glorieta) were evaluated. In all almond cultivars, the roasting process enhanced both antioxidant activities and bioactive compounds. In particular, roasting had a very large effect on the total phenolic content of the Refêgo cultivar, increasing it from 0.02 to 2.66 mg gallic acid equivalent/g fresh weight. Antioxidant activity in Refêgo cv after roasting was accompanied by a 69% increase in ABTS activity. This increased antioxidant activity could be due to roasting inducing cell wall disruption, allowing better antioxidant extraction. The authors also hypothesized that chemical modifications may be induced as a consequence of heat. Additionally, they observed that after roasting, there was an increase in polyunsaturated fatty acids, whereas saturated and monounsaturated fatty acids decreased. The 'cultivar' effect in these variations was important. These findings are in disagreement with those of Valdés et al. [70], who found that the contents of saturated fatty acids and monounsaturated fatty acids increased while that of polyunsaturated fatty acids decreased in almonds after processing. The different results obtained by the two authors could be due not only to the 'cultivar' effect but also to the different processing conditions as well as the fact that minor fatty acids are often

not measured. Among investigated sensory attributes, those that differentiated the almond cultivars after roasting were found to be skin color, bitter almond flavor, bitter taste and sweet almond flavor. Roasting positively affected the perception of skin color and sweet almond flavor in Ferragnès, while bitter almond flavor and bitter taste decreased in Molar cv. Generally, the roasting led to kernels with a strong, sweet almond flavor. More recently, Caltagirone, Peano and Sottile [71] studied the influence of some post-harvest industrial processes on the nutraceutical properties of three different Italian almond cultivars (Tuono, Genco and Vinci a tutti). The authors highlighted that for all cultivars, the roasting of almonds yielded both the highest antioxidant capacity and phenolic compound content. After roasting, total polyphenol content was increased in the Vinci a tutti cv (33.6 mg/g), followed by Genco cv (32.1 mg/g) and Tuono cv (31.4 mg/g). The work confirmed the positive relationship between phenolic compound content and antioxidant capacity after the roasting process, so far also found in other nuts such as chestnuts [72]. Nevertheless, discrepancies among data on phenolic content after roasting are reported in the literature. Some studies, indeed, showed a decrease in phenolic content immediately after roasting [73]; others reported a notable increase influenced by separation methods [74]. Sruthi, Premjit, Pandiselvam, Kothakota and Ramesh [75] hypothesized that this behavior probably can be interpreted as an increase in the extractable phenolic compounds post roasting. Garrido et al. [74] reported that, in addition to polyphenols, 5-hydroxymethyl-2-furaldehyde was found in high concentrations in blanched and especially in roasted almonds. This compound, a derivate from the Maillard reaction following thermal treatments, shows antioxidant properties able to enhance the total antioxidant capacity of almonds after processing, such as roasting and blanching. On the other hand, the blanching treatment is a thermal process in which almond skins are removed [15] to reduce potential contamination, such as bacterial and mold growth. Regarding the effect of blanching treatment on almond characteristics, it is important to underline that phenolic compounds are present mainly in the skin of almonds. In these cases, certain nutritional characteristics are, therefore, lost due to the removal of the skin from the almond [74]. The effects of the blanching process on almond bioactive components have been reported by Oliveira et al. [69], who demonstrated that both bioactive compounds and antioxidant activities are reduced after blanching. The authors reported that the effect of the blanching process on phenolic compounds differed according to the cultivar; in fact, no significant effect was found for Casanova, Ferragnès, and Glorieta cv. In Pegarinhos cv, the phenolic content decreased from 0.19 in raw almonds to 0.08 in blanched samples (mg gallic acid equivalent/g fresh weight). Regarding antioxidant activities in all samples after blanching, enormous decreases were found. In blanched Pegarinhos almonds, ABTS activity was around 17 times lower than that in the raw almonds (0.68 versus 11.59 μg Trolox/g).

## 6. Sensory Analysis

The importance of sensory analysis in evaluating the quality of almonds has been recognized and demonstrated by the many types of research carried out [76–82]. In 2010, Civille et al. [77] developed a lexicon to describe the almond sensory profile. In this study, the authors analyzed 20 samples from Californian regions over two harvest years. This interesting work included the seven most representative varieties of almonds (Nonpareil, Carmel, Butte, Monterey, Fritz, Mission and Sonora). Panelists drafted a list of descriptive terms to characterize the appearance, aroma, flavor and texture attributes of the almonds. The results indicated that each variety had unique attributes. Another useful finding was that all samples presented modest color intensity, as well as flavor attributes with intensities that ranged between 0 and 5, indicating the presence of mild flavor. These findings also showed that sensory analysis could help to distinguish different varieties of almonds in terms of appearance, aroma, flavor and texture attributes. Further investigation of the sensory differences between almond cultivars was carried out by Contador et al. [79], who highlighted the importance of texture parameters in evaluating the sensory quality of raw almonds. The 'Nonpareil', 'Mission', 'Supernova', 'Tuono', 'Ferragnès' and 'Marcona'

cultivars from Chile were analyzed. Color intensity, roughness, flavor intensity, crispness, crunchiness and hardness were the attributes considered by a trained panel of 14 assessors. A scale ranging from 0 to 15 was utilized. The authors reached interesting conclusions: the panelists, analyzing all studied cultivars, identified 'Supernova' and 'Tuono' as the tastiest and 'Tuono' as also the crunchiest. At the same time, the American cultivars 'Mission' and 'Nonpareil' appeared to be different from the European cultivars 'Marcona', 'Supernova', 'Tuono' and 'Ferragnès' in their sensory characteristics. The influence of different packaging, temperatures and storage times on wild almonds was evaluated with consideration not only for their oxidative stability but also their sensory properties [80]. The authors assessed the influence of temperature (4, 25 and 35 °C) and atmosphere (vacuum, $CO_2$ and normal air) on almond quality. Sensory evaluation of samples, grown in Southern mountainous areas of Iran, was performed by 30 judges. The parameters of odor, flavor, color, juiciness and overall acceptance were evaluated using a hedonic scale (1 = extremely dislike to 5 = extremely like). The sensory analysis in all of the compared temperatures indicated that the overall acceptability of all packaged samples decreased through the storage period. As expected, the increase in storage temperature resulted in a decrease in the overall acceptability of all samples with lower sensory scores. In more current research [81], both descriptive and affective sensory analyses of four almond cv growing under different irrigation types were conducted to study their acceptance and consumer motivation to pay for hydro-sustainable almonds. Fruits and vegetables cultivated under controlled deficit irrigation are called hydro-sustainable (hydroSOS) products since they are environmentally friendly. The descriptive analysis was conducted by 10 trained panelists who defined a lexicon list of descriptors. Then, they analyzed almond samples to define each attribute's intensity using a scale (from 0 to 10). The findings were not different among treatments both for descriptive and affective sensory analysis. A particularly important aspect was that consumers were willing to pay a higher price for hydroSOS almonds. This fact demonstrates the attitudes and growing attention of consumers toward sustainable foods.

## 7. Quality of Almonds Valued by Non-Destructive Analysis

Recently, researchers demonstrated the importance of near-infrared spectroscopy (NIR) as a valid, non-destructive method for analyzing food products [83–85]. NIR spectroscopy is actually considered an approach very useful for evaluating the qualitative characteristics of almonds, too [86–91]. Arndt et al. [86] investigated four different sample preparation techniques for near-infrared (NIR) spectroscopy analysis of almonds, evaluating their suitability for determination of geographical origin. In this research, 64 almonds from different countries (Australia, Spain, Morocco, Italy, Iran and the United States) were analyzed as whole, bisected nuts and in ground and freeze-dried states (after grinding). Reported data showed that the freeze-dried almonds had the highest classification accuracy of 80.2%. This evidence was due to the removal of water, which produced a signal overlay that resulted in information loss. The analysis of ground and bisected almonds yielded accuracies of 71.9% and 64.5%, respectively, followed by the analysis of whole almonds with an accuracy of 62.6%. In this case, the reduced percentage accuracy of whole almonds could be due to the influence of the tegument. The authors concluded that the most promising sample preparation technique for determining the geographical origin of almonds was found to be freeze-drying after grinding, though the most easily and rapidly feasible analysis for an initial screening remained the whole almond. These findings were later confirmed by Arndt et al. [87], who reported the prediction of the geographical origin of almonds (Prunus dulcis MILL.) via Fourier transform near-infrared (FT-NIR) spectroscopy for 250 almond samples from the same six countries (Australia, Spain, Morocco, Italy, Iran and the United States) analyzed in the previous work [85]. A support vector machine model facilitated a mean classification accuracy of 80.3%, and the distinction between Mediterranean almonds and American almonds was possible to recognize. Additionally, by combining the Italian and Spanish almonds into one

Mediterranean class, an accuracy of 88.2% was obtained. It is important to underline that a limitation of this model is the analysis of almonds from other countries, as it did not allow allocation to an unknown class. All considered, it can be concluded that NIR screening is suitable for the determination of the geographical origin of almonds. In another study [89], the fatty acid profile of 149 samples of shelled sweet and bitter almonds was measured using a line-scan hyperspectral reflectance imaging system working in the NIR (946.6–1648.0 nm) range. The authors analyzed 89 samples of sweet almonds (Prunus dulcis Mill., cv. Antoneta, Belona, Guara, Lauranne, Soleta, and Vairon) and 60 samples of bitter almonds of non-specific cultivars. The applied hyperspectral imaging calibration models had the best performance when quantifying oleic and linoleic acids; regarding the rest of the fatty acids analyzed and the oleic to linoleic ratio, the models could be used for screening. The obtained data demonstrated that the hyperspectral imaging system can be considered an emerging approach for estimating fatty acids in almonds. Vega-Castellote, et al. [90] showed that the use of near-infrared spectroscopy (NIRS) represents a rapid and sure method for the determination of amygdalin amounts and for almond sorting based on bitterness. Using NIRS methods, it was possible to precisely detect bitter almonds, both in-shell and shelled. This evidence showed that the NIRS technology is also suitable for use in the industrial control of the amount of amygdalin in almonds, which is a benefit for the almond industry compared to other methods normally used. In addition, an innovative system based on NIR spectral information was implemented for detection of non-compliant batches of sweet almonds [91]. A total of 140 samples of shelled almonds, 90 sweet varieties (Antoñeta, Belona, Guara, Lauranne, Soleta and Vairon cvs) and 50 non-specific bitter cvs were analyzed. The development of NIRS technology allowed for the detection of sweet almond batches adulterated with bitter almonds, achieving a detection rate of 87%. This approach could allow for operating analyses of almonds during the industrial process. These findings confirmed the suitability of NIR screening for the determination of almond quality and may pave the way for future analytical applications.

## 8. Conclusions

This literature review revealed that many studies on almonds were, above all, concentrated on processing and storage methods to extend shelf life, health benefits and nutritional value. We found that fewer articles in the literature had been published on the investigation of topics such as sensory analysis. Future research could focus on the sensorial traits of the various almond cultivars and the characterization of their sensory features. The ultimate goal should be to monitor the quality of almonds using an official panel to allow for more efficient sensory analysis.

**Author Contributions:** All authors contributed to manuscript writing. Conceptualization, review and editing, R.M. and M.T.F. All authors have read and agreed to the published version of the manuscript.

**Funding:** This research received no external funding.

**Institutional Review Board Statement:** Not applicable for studies not involving humans or animals.

**Informed Consent Statement:** Not applicable for studies not involving humans.

**Data Availability Statement:** The data presented in this study are available on request from the co-author, Maria Teresa Frangipane.

**Acknowledgments:** The authors gratefully acknowledge the 'Departments of excellence 2018' program (i.e., 'Dipartimenti di eccellenza') of the Italian Ministry of Education, University and Research for the financial support through the 'Landscape 4.0 food, wellbeing and environment' (DIBAF department of University of Tuscia, Italy).

**Conflicts of Interest:** The authors declare no conflict of interest.

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
