# Peer review of "Progress in Almond Quality and Sensory Assessment: An Overview"

_agriculture, doi:10.3390/agriculture12050710_

Round 1

Reviewer 1 Report

Many of the changes recommended were ignored and was difficult to determine what changes had been made in the revised manuscript.

Line 11 make as estimate 3.2 million

Abstract contains some hyphens that are unnecessary within words

Line 59 Prunus dulcis needs to be italicized

Line 84 convert to approx. tonnes

Line 99 needs to be approx.

Lines 109-110 How does this review help the academic research community rather than producers?  Needs to be described.

Still no ordering of section 3 into separate paragraphs which would help the reader.  Also with other sections too.  It appears that no changes have been made to improve readability – could organize into separate paragraph for each nutrient and explain its importance.

Line 26 sentence has not been changed

Line 150 the description of only two breeds could be described earlier before describing the large genetic variability in line 137

Line 615 what is meant by adulterated?

Author Response

Many of the changes recommended were ignored and was difficult to determine what changes had been made in the revised manuscript.  

The authors’ response to the reviewers’ comments is highlighted in yellow as well as with yellow evidence all other corrections were done. We would like to thank the reviewer for providing the valuable comments we received, very much appreciated. We agree with most of them.

Reviewer 2 Report

The article was innovative in summarizing the fruit characteristics, nutritional value, and the effect of processing on shelf life. However, the content needs more of a distillation of the main idea rather than a stacking of information. This is a huge work that needs to be carefully revised and summarized.

When studying almond flavor substances, it is recommended to focus on adding the content related to bitter components. The varietal characteristics and differences between bitter almonds and sweet almonds, the flavor differences and the reasons for their generation, and the differences in processing, are what I think need to be discussed in a focused manner. Further, the changing pattern of D (-)-Amygdalin hydrate in bitter almonds and the reasons for its generation should also be important details for discussion, considering that D (-)-Amygdalin hydrate tastes bitter, toxic, and may not have anti-cancer effects.

Almond is nutritious and unique in taste, and it is recommended to supplement its commercial development and product variety.

This research mentioned the use of color, volatile compounds, sensory evaluation, and nondestructive analysis in almond quality discrimination. However, the section is not well summarized in the abstract, preface, and conclusion, nor is it analyzed in the body in a very reasonable and logical manner.

Author Response

We would like to thank the reviewer for providing the valuable comments we received, very much appreciated. Most of the lacking was modified and we hope that our review is also an important resource for scientists. It could offer inspiration for one’s own research to other researchers.

Reviewer 3 Report

Most of lacking were modified.

Author Response

We would like to thank the reviewer for providing the valuable comments we received, very much appreciated.

Reviewer 4 Report

These authors provide an overview in order to improve the quality of almonds and the sustainability of the whole production. The quality attributes of almonds, in terms of their nutritional characteristics, health benefits, and the impact of processing on shelf life were investigated. However, the paper needs slightly improvement. My detailed comments are as follows:

  1. Page 6, Line 219: “200.6g/kg-1”, delete “g/kg-1”.
  2. Please specify whether the active ingredient and antioxidant capacity content is on a dry or wet basis.

Author Response

The authors’ response to the reviewers’ comments is highlighted in yellow as well as with yellow evidence all other corrections were done. We would like to thank the reviewer for providing the valuable comments we received, very much appreciated.

Round 2

Reviewer 1 Report

There are a few typos and requirement for the iltalicizing genus names - please read through

Line 217 and 228 about minerals being effected by genotype/ environmental factors - contradict each other

Author Response

We would like to thank the reviewer once again for providing the valuable comments we have received, much appreciated.

Reviewer 2 Report

D-Amygdalin of almond of Prunus dulcis needs to be discussed, which is mentioned in Line 468/668/672. The amount of amygdalin in almond depends on the growing conditions of the tree. High levels of amygdalin can cause poisoning when untreated almond, especially bitter ones, are consumed. For adults, ingestion of 50 bitter almond can lead to severe poisoning or even fatal. For children, the lethal dose is about 5-10 capsules. In traditional medicine, bitter almond aqua were used as sedatives, tonic and pain relievers. Historically, both bitter and sweet almond were used to make syrups, but because of the amygdalin content of bitter barberry kernels, syrups now marketed are usually made from sweet barberry only.

In addition, it is recommended to add the description of almond (Prunus dulcis) oil. Bitter almond is not edible, but the oil extracted from them is distilled to remove the toxic substance amygdalin. Almond oil can be used in the production of soap and can also provide a unique bitter almond flavor to dishes and beverages. Suggested reference article: Slavica ÄŒolić, Gordan Zec, Maja Natić, Milica Fotirić-Akšić, Fruit Oils: Chemistry and Functionality, 2019, ISBN : 978-3-030-12472-4, Sci-Hub | Almond (Prunus dulcis) oil. Fruit Oils: Chemistry and Functionality, 149–180 | 10.1007/978-3-030-12473-1_6.

Author Response

We would like to thank the reviewer once again for providing the valuable comments we have received, much appreciated.

This manuscript is a resubmission of an earlier submission. The following is a list of the peer review reports and author responses from that submission.

Round 1

Reviewer 1 Report

1. Authors provide many different news and stories in this artilce, even tell us about the myth about almond. However, for a scientic review article, it is too cumbersome and not easy for readers to get  a complete discussion. 

2.Authors did not let this review to hlep readers to get the information of " This review will help almond producers to select the best varieties to grow in a specific geographical region and ultimately improve the quality of almonds produced."  which authors mentioned in line 19-20.

3.Authors hope we can get the information about  " how to select the best varieties to grow in a specific geographical region" , but we can not get the discussion in the manuscript. Even there are no information about how enviroment effect the quality of chemical composition and nutritional value on almondfruit.

4. Authors also hope this manuscript can help to "Ultimately improve the quality of the almonds produced", however, we couldn't get enough information from this article. About this,  we can see from conclusion that already clearly showed the reference were not enough and related research are lacking. So I think though this topic is important but there are still not enough data or information in this field to complete a suitable review artilce. Authors need to find more reference to support their surpose and imporve the writing to provide more clearly and convincing paper.

Author Response

We would like to thank the reviewer for providing the valuable comments we received, very much appreciated. We agree with most of them.

Reviewer 2 Report

The article was innovative in summarizing the current status of research from the perspectives of fruit characteristics, nutritional value, and sensory evaluation. However, the review was poorly summarized, with redundant statements throughout and a low level of scholarship and expertise in the latest research developments. The content of the manuscript need to have more scholarly research arguments rather than a pile of unfocused information. This is a huge undertaking that needs to be carefully revised and summarized.

  1. The discussion in the preface is unfocused.
  2.   In “Almond fruit characteristics”, the description of fruit characteristics could be considered in the form of a table summarizing the domestication history of different varieties and origins of almonds, as well as the characteristics of shape, growing period, and yield.
  3. A summary of the bioactive substances and medical perspectives of almonds is best presented in tabular form as well.
  4. When studying almond flavor substances, it is recommended to focus on the addition of bitterness-related content.
  5.  Almonds are nutritious and unique in taste, and it is recommended to supplement its commercial development and product variety.
  6.  It is recommended to keep the index of almond sensory evaluation, but the sensory evaluation method does not need to be written more.

Author Response

(The authors gave the same response as above.)

Reviewer 3 Report

The review describes the history of almonds and a myth about the origins of almonds.  Almonds are derived from two breeds but has one of the largest genetic variabilities of any fruit.  There is a large description of the composition on varieties showing high biomass yields, protein, lipid and mineral contents which as expected can show significant variation.  Almonds contain considerable quantities of antioxidants especially in certain varieties and their health benefits are described in lowering cholesterol and diabetes.  The effects of processing is described in terms of colour development and taste.  Research on sensory taste of almonds and its crunchiness is described as well as new methods to quickly analyse the almond composition using NIR.

Perhaps what is lacking in this article is an analysis of the collected data to provide something new

In abstract: units megagrams are confusing – why not use tonnes? There is very little in the article about shelf-life and sensory studies have been omitted

Section 3 is hard to follow and some order is required

section of domestication of almonds could go earlier after describing origin of almonds [4] in order

to maintain a timeline

Fig. 1 provide more description about times of year of trees

Line 102 change to cycle of the almond

Line 126  45% could be saved by using less water during the latter stages - needs to be stated again.  Also begin new paragraph at Almonds….

Further paragraphs are required within this text e.g. line 135

Line 150 the description of only two breeds could be described earlier before describing the large genetic variability in line 137

Line 446 what are L, a and b values?

Line 451 change to findings agreed with Kaften [63] indicating both pigment

line 461 this sentence could be changed

line 466 change more to greatly

Line 469  the increase in minerals could be caused by drying whereby water content is decreased during roasting

Line 474 do not understand the logic why lower antinutrients

Line 479 can the degradation of oxalates be described further?

Line 487 very large effect

Line 488 increased from

Line 509 these values are unlikely to be significantly different

Line 530 could help

Line 552 More current research defined

Line 555 what is hydroSOStainable?

Line 571 64 almond varieties?

Line 615 what is meant by adulterated?

Author Response

(The authors gave the same response as above.)

Reviewer 4 Report

These authors provide an overview in order to improve the quality of almonds and the sustainability of the whole production. The quality attributes of almonds, in terms of their nutritional characteristics, health benefits, and the impact of processing on shelf life were investigated. However, the paper needs slightly improvement before acceptance for publication. My detailed comments are as follows:

1. It is recommended that a detailed description of the nutritional value of almonds be added to the abstract section.

2. There is no denying that processing methods are critical to the quality of almonds. The authors extensively discuss the effect of roasting on almond quality, but the discussion of the effect of processing methods such as hulling, boiling and blanching on almond quality is very limited. It is recommended to add relevant content to complete the fifth chapter section.

3. Phenols are not the only compounds that determine the antioxidant capacity of almonds. This is because chemical reactions that occur during processing can also produce new chemical components that enhance the antioxidant capacity of foods, such as the Merad reaction. It is recommended to add related content.

4. Page 6, Line 219: “200.6g/kg-1”, please check.

5. Page 7, Line 269: “679.53”, please add the unit.

6. Page 9, Line 319-321: “There is a growing interest in almond antioxidant compounds because of their multiple functions, antioxidant and nutraceutical properties and their potential to extend the shelf life of almonds.” Please provide the relevant reference to support this argument.

7. Page 9, Line 340: Delete “period of”.

8. Page 13, Line 496-498: Please explain why the findings of the two authors differ.

9. For the experimental results of the determination of active ingredients and antioxidant capacity, are they calculated by dry weight or wet weight?

Author Response

(The authors gave the same response as above.)
